# Recent Tendency on Transition-Metal Phosphide Electrocatalysts for the Hydrogen Evolution Reaction in Alkaline Media

**DOI:** 10.3390/nano13182613

**Published:** 2023-09-21

**Authors:** Seo Jeong Yoon, Se Jung Lee, Min Hui Kim, Hui Ae Park, Hyo Seon Kang, Seo-Yoon Bae, In-Yup Jeon

**Affiliations:** Department of Chemical Engineering, Nanoscale Environmental Sciences and Technology Institute, Wonkwang University, 460 Iksandae-ro, Iksan 54538, Jeonbuk, Republic of Korea; dbstjwjd1210@wku.ac.kr (S.J.Y.); lse1222@wku.ac.kr (S.J.L.); kppo112@wku.ac.kr (M.H.K.); alicia3207@wku.ac.kr (H.A.P.); hyoseon0131@wku.ac.kr (H.S.K.)

**Keywords:** hydrogen evolution, transition metal phosphides, electrocatalysts

## Abstract

Hydrogen energy is regarded as an auspicious future substitute to replace fossil fuels, due to its environmentally friendly characteristics and high energy density. In the pursuit of clean hydrogen production, there has been a significant focus on the advancement of effective electrocatalysts for the process of water splitting. Although noble metals like Pt, Ru, Pd and Ir are superb electrocatalysts for the hydrogen evolution reaction (HER), they have limitations for large-scale applications, mainly high cost and low abundance. As a result, non-precious transition metals have emerged as promising candidates to replace their more expensive counterparts in various applications. This review focuses on recently developed transition metal phosphides (TMPs) electrocatalysts for the HER in alkaline media due to the cooperative effect between the phosphorus and transition metals. Finally, we discuss the challenges of TMPs for HER.

## 1. Introduction

Given the challenges posed by rising global energy demand and the associated climate changes attributed to fossil fuel consumption, it has become imperative to explore sustainable and environmentally friendly alternative energy sources, thereby reducing our reliance on fossil fuels [1]. Amidst the array of clean energy sources, hydrogen is magnified as a particularly crucial and promising contender. This clean energy source has garnered substantial interest within the realm of renewable energy due to its environmentally friendly characteristics, absence of emissions, and an impressive gravimetric caloric value reaching 120 MJ/kg [2]. According to the International Energy Agency and the Hydrogen Council, more than 500 Mt of global hydrogen will be required and about USD $3 trillion in revenue generated through the hydrogen-value chains in 2050 [3]. So, it is important to efficiently produce hydrogen for sustainable development. At present, the predominant method for producing hydrogen at a large scale involves steam reforming of fossil fuels. However, this approach worsens the depletion of finite fossil resources and contributes to a higher carbon footprint [4]. In stark contrast, water electrolysis stands out as an environmentally viable and sustainable alternative. This method harnesses the advantages of using water as a raw material and avoids the emission of greenhouse gases, making it an attractive option for hydrogen production [5,6,7,8,9].

Indeed, the advancement of efficient electrocatalysts for water splitting is crucial for the large-scale manufacture of H_2_ gas and its commercialization as a clean energy source. Historically, precious metals like platinum (Pt), ruthenium (Ru), palladium (Pd), iridium (Ir), and others have been highly regarded for their exceptional electrocatalytic properties in facilitating the hydrogen evolution reaction (HER) [10,11,12]. Nonetheless, the extensive adoption of these noble metals for electrocatalysis is hindered by their significant drawbacks, primarily the substantial cost and limited availability. As a result, the activation and optimization of non-noble transition-metal electrocatalysts for HER presents itself as a viable avenue towards achieving cost-effective hydrogen production [13]. Researchers have directed their efforts towards creating electrocatalysts for HER using elements that are plentiful in the Earth’s crust. This involves a notable emphasis on economically viable transition metals like iron (Fe), cobalt (Co), nickel (Ni), molybdenum (Mo), tungsten (W), and non-metal elements like carbon (C), oxygen (O), nitrogen (N), phosphorus (P), and sulfur (S). These materials have undergone extensive investigation in alkaline environments as potential replacements for precious noble metal catalysts [14]. Especially in alkaline media, HER catalysts using abundant elements actively have been intensively developed because the alkaline medium serves as a promising platform for materials that might not be as appealing in acidic conditions due to their limited stability. For instance, catalyst materials like pure transition metals [15], alloys [16,17,18], oxides [19,20], and hydroxides [21] often face challenges in maintaining stability within acidic environments. However, the alkaline medium provides a conducive environment for these materials to exhibit their catalytic prowess [22]. Therefore, enhancing the HER electrocatalytic performance in alkaline media is particularly significant to developing water electrolysis for hydrogen production [23]. Furthermore, the utilization of the alkaline environment for the HER opens the door to compelling replacements that enhance electrocatalyst stability and cost-effectiveness. This is achieved by opting for non-precious transition metals as electrocatalysts, leading to improved sustainability and reduced expenses [14,24,25,26,27,28].

Transition-metal phosphides (TMPs) have earned significant interest as auspicious catalysts for HER due to their catalytic mechanism resembling that of hydrogenases. This resemblance underscores their potential advantages in driving efficient HER processes [29,30,31,32,33,34]. Typically, hydrogen exhibits a strong binding to TMPs, prompting extensive research into doping cations or anions as an efficient strategy to adjust their electronic properties and optimize the free energy of hydrogen adsorption (ΔG_H*_) to further improve their performance in the HER [35,36,37,38]. TMPs have garnered interest as electrocatalysts primarily because of the synergistic fusion of their distinctive structural characteristics [39]. Owing to the presence of metal–metal bonding networks, these characteristic TMPs demonstrate metallic conductivity when certain combinations of metals and phosphorus are present. This inherent property is critical for facilitating the development of high-performance electrocatalysts [40,41]. The elevated electronegativity of phosphorus (P) atoms in TMPs enables them to effectively attract electrons from metal atoms, facilitated by the negatively charged nature of these phosphorus atoms. Conversely, the P atoms can also play a “base” role by capturing charged protons during the course of electrocatalytic reactions [42]. To clarify, in the case of TMPs, the high electronegativity of phosphorus (P) atoms results in negatively charged P atoms, while the metal atoms become positively charged. These charged P atoms and metal atoms play as centers that can accept protons and hydrides, respectively. This dual role leads to a cooperative effect that enhances the efficiency of HER [43]. The negative charge on the P atom, which functions as a proton acceptor, serves to weaken the strength of the metal bonds. This weakening effect facilitates the desorption of hydrogen [44,45]. The moderate bonding between phosphorus and the reaction intermediates plays an important role in preventing the slow desorption of H_2_. This is in contrast to pure metals, where considerably stronger adsorption can lead to sluggish H_2_ desorption [46,47,48,49].

This review summarizes the recent TMP electrocatalysts for HER. It briefly introduces the HER mechanism for scientific understanding and the preparation of TMPs. Then, the HER performance of TMPs with phosphorus (P) and transition metals (e.g., Ni, Co, Cu, Fe, Mo, etc.) are shown. Finally, this review discusses the challenges of TMPs as electrocatalysts for HER.

## 2. HER Mechanism

The HER, constituting the cathodic half-reaction of water splitting, facilitates the generation of H_2_ gas by reducing protons and water molecules (Figure 1). This process is elucidated by the equations detailed in Table 1. Typically, the HER follows a multi-step pathway, often involving the Volmer–Heyrovsky or Volmer–Tafel mechanisms, regardless of the medium. Irrespective of the specific pathway, the reaction consistently progresses through the adsorption of hydrogen intermediates (H_ads_) on the surface of catalysts [50,51].

In alkaline media, the concentration of protons in the electrolyte is exceedingly minimal; therefore, the initial Volmer step is crucial for reducing H_2_O and dissociating O-H bonds to form H_ads_ on the active sites of the catalytic surface [52,53]. This marks the initial stage of the HER, referred to as the Volmer reaction. During this process, transferred electrons have the capability to generate adsorbed hydrogen species (H_ads_) alongside negatively charged hydroxide anions. During the subsequent phase of the HER, the second step involves the potential production of gaseous hydrogen, which can materialize through either the Heyrovsky or Tafel pathways. The Heyrovsky reaction is as follows. H_ads_ binds to another water molecule and an electron to produce a hydrogen molecule and a hydroxide anion [54]. The Tafel reaction combines two H_ads_ atoms regardless of the pH in the media [55]. In theory, the Tafel slope can be used to evaluate the dominant reaction mechanism in the HER process [1]. The smaller the Tafel slope, the faster HER electrocatalytic kinetics, and the HER rate-determining step (RDS) can be identified from the Tafel slope [56].

## 3. Preparation of Transition-Metal Phosphides (TMPs)

The synthesis process of TMPs is significantly impacted by the reaction conditions and precursors employed, ultimately exerting a critical influence on the ensuing electrocatalytic performance. This impact is achieved by controlling the composition, crystal phase, and overall structure of the resulting catalysts [57]. As a result, various physical and chemical approaches have been developed to augment the performance of diverse nanomaterials (Figure 2). The chosen synthesis approaches play a pivotal role in determining the characteristics of TMPs, including factors like size, morphology, and the distribution of P atoms within the structure [58].

### 3.1. Solution-Phase Reaction

Tri-n-octylphosphine (TOP), trioctylphosphineoxide (TOPO), and triphenylphosphine (PPh_3_) are three common phosphorus sources that have been extensively utilized to synthesize transition metal phosphides (TMPs) through solution-phase reactions [59,60]. The P atom can be readily liberated through the breaking of the covalent C-P bond within the P source molecule [61]. Furthermore, the utilization of P source materials can expedite the reaction process and result in the formation of unconventional structures [29]. Nonetheless, it should be noted that rigorous air-free operations are required to avoid highly flammable P at high-reaction temperatures. While the synthesis of metal phosphides with controllable sizes and defined nanostructures can be achieved through the use of organophosphorus compounds, the advancement of this synthetic approach is constrained due to its complexity. This process involves pyrophoric and toxic organic phosphine, which poses challenges for further development [62]. For example, Xu et al. reported the nanoporous FeP nanosheets via an anion exchange reaction [63].

Until now, hypophosphite (e.g., NaH_2_PO_2_ and NH_4_H_2_PO_2_) has been frequently utilized as an alternative P precursor [64]. As an inorganic P source, hypophosphite can decompose in situ to generate gaseous PH_3_. The in situ generated PH_3_ can react with various metal precursors while maintaining the dimension and structure of the precursors.

### 3.2. Solid-Phase Reaction

The solid-phase reaction involves the combination of solid metal and phosphorus sources, succeeded by thermal treatment in an inert or vacuum conditions [65]. Solid metal sources encompassing metal nanoparticles, metal oxides, or metal hydroxides, as well as phosphorus sources like red phosphorus or di-ammonium phosphate, have been employed. By altering the metal precursors and adjusting reaction temperatures, it is possible to produce various crystalline phases of TMPs. Each of these phases is distinguished by uniform distributions of shapes and sizes [58]. Consequently, adjusting the reaction temperature and reactant molar ratio offers the ability to optimize the performance of TMPs [66]. Xiao et al. reported MoP for the electrocatalyst using this method [67].

### 3.3. Gas-Solid Phase Reaction

Pre-synthesized solid-state precursors can be effectively transformed into TMPs using the gas–solid phase reaction. The use of PH_3_ gas proves to be efficient in phosphorization, reacting with various metal sources like bulk metal, metal oxides, metal–organic complexes, and metal hydroxides, leading to the formation of TMPs at different temperatures [58]. Beyond the phosphorization of pre-synthesized precursors, substrates composed of transition metals, like nickel foam [68], nickel foil [69] and iron foam [70], can undergo direct phosphorization via the gas–solid phase reaction to generate TMPs [52]. The common approach for the gas–solid phase reaction often utilizes a surfactant-free process that helps maintain the surface morphology and dimensions of the precursor materials. For example, Geng et al. synthesized the hole-rich CoP nanosheets through the gas–solid phase reaction [71].

### 3.4. Decomposition

The decomposition of a metal–organic precursor is a commonly employed method, executed at moderate temperatures. This approach offers the advantage of mitigating challenges related to the handling and storage of highly pyrophoric reagents like phosphine gas or white phosphorus. In this process, a combination of metal–organic and organic phosphorus compounds are blended, leading to the formation of a metal–organic precursor. Subsequent thermal decomposition results in the ultimate formation of TMPs. By adjusting the temperature during the phosphidation reaction, various crystalline phases of TMPs can be obtained. Additionally, manipulation of the metal (M)/phosphide (P) ratios is achievable by modifying the proportions of the metal and TOP [58].

TMPs can be synthesized via the pyrolysis of precursors that encompass metal, phosphorus, and carbon components [72]. Notably, phytic acid (PA, C_6_H_6_(H_2_PO_4_)_6_), which comprises six phosphonic acid groups, holds substantial appeal as a phosphorus source owing to its robust coordination capability and elevated phosphorus content. The six phosphoryl groups present in PA make it an efficient crosslinker with metal ions, leading to the formation of metal–PA complexes. These complexes, containing both metal and phosphorus, undergo carbonization to yield the final TMPs [51,73].

### 3.5. Hydrothermal (or Solvothermal) Reaction

The hydrothermal (or solvothermal) reaction, a prominent bottom-up synthetic technique, serves as a prevalent method for producing nanomaterials. It involves the thorough mixing of specific chemicals in distilled water and subsequent sealing within an autoclave [52]. Hydrothermal processes predominantly utilize water as the solvent, while solvothermal methods involve the use of organic solvents to synthesize TMPs. Precise quantities of metal and phosphorus sources are dissolved in an appropriate volume of distilled water or an alternative solvent. The resultant mixture is then moved to a sealed stainless steel (or Teflon-lined) autoclave. Hydrothermal reactions are conducted at diverse temperatures (ranging from 120 to 200 °C) and durations [58]. The elevated temperature and pressure within the autoclave facilitate the expedited arrangement of various nanostructures [74,75]. Changes in the temperature, time, and concentration of the precursor materials can lead to different sizes and morphologies of TMPs. Due to the mild reaction conditions, the hydrothermal reaction has a greater possibility of achieving the economic production of heterostructure materials on a large scale [76]. Das et al. synthesized a binary metal phosphide (NiCoP) using the hydrothermal method [77].

### 3.6. Electrochemical Deposition

Electrochemical deposition stands out as an appealing technique for crafting heterostructure electrocatalysts intended for the HER. This method is prized for its adaptability and mild reaction conditions, effectively sidestepping the necessity for high pressure, elevated temperatures, and environmentally harmful gases [76]. The precursors generated through the electrochemical deposition showcase diverse compositions and morphologies, which can be subsequently phosphorized to yield TMPs. This technique accommodates a wide array of compositions and affords nanometer-level precision for modifying crystal growth—a level of precision that proves challenging to attain through alternative means. Moreover, electrochemical deposition can be harnessed for the creation of bimetallic phosphide precursors [52]. This versatile approach offers the potential to fine-tune the inherent activity of heterostructure catalysts designed for HER [76]. Lu et al. developed an electrodepositions process to synthesize FeP film for electrocatalysts [78].

### 3.7. Electroreduction

Electroreduction is a graceful method for phosphorization using metal and hypophosphite ions. Under reducing potentials, H_2_PO_2_^−^ is converted into PH_3_, which is then accumulated with metal ions to make TMPs [79]. This approach necessitates mild conditions, rendering it environmentally friendly and efficient, aligning with the principles of green and straightforward TMP fabrication. Zhu et al. studied Ni-based phosphide fabricated by the electroreduction method [80].

## 4. Electrocatalytic Performance for HER in Alkaline

### 4.1. Ni-Based TMPs

#### 4.1.1. Ni-P Structure

Nickel phosphides have eight mono- or poly-phosphides with different Ni/P ratios, and the stoichiometric compositions, crystal phases, and crystal facets affect electrocatalytic activity. In other words, nickel phosphides that have unique electronic properties and excellent anti-corrosion properties have been widely studied as an electrocatalyst [81].

Shi et al. introduced a novel Ni_2_P nanosheets/Ni foam (Ni_2_P/Ni) via a straightforward chemical conversion approach. They utilized a surface-oxidized Ni foam as a precursor and employed a low concentration of trioctylphosphine (TOP) as the phosphorus source [82]. Intriguingly, the recorded overpotential at a current density of 10 mA cm^−2^ was approximately 41 mV, with a calculated Tafel slope of 50 mV dec^−1^. Additionally, the Ni_2_P/Ni composite demonstrated remarkable stability within an alkaline environment.

Yan et al. presented a study involving nickel-based metal–organic frameworks (MOF-74-Ni) integrated with reduced graphene oxide (rGO) to create a Ni_2_P/rGO composite [83]. Through a one-step calcination process at low temperature, using sodium hypophosphite as the phosphorus source, they obtained Ni_2_P/rGO. This composite exhibited significant merits, including a substantial active surface area, optimal distribution of active sites with ultra-small particle dimensions, and evident defects on the exposed rGO sheets. These defects indicated the formation of ultra-small nickel phosphide nanocrystals on both sides of the rGO, resulting in a sandwich-like Ni_2_P/rGO structure. The Ni_2_P/rGO composite displayed a notably low overpotential of 142 mV under 10 mA cm^−2^. Noteworthy Tafel slope data were observed, registering at 58 mV dec^−1^ for the Ni_2_P/rGO with a calculated exchange current density of 3.1 × 10^−5^ A cm^−2^. Impressively, the Ni_2_P/rGO demonstrated exceptional endurance within an alkaline medium, evident from a stable chronopotentiometric curve sustained over 20 h.

#### 4.1.2. Ni-N&P Structure

Jin et al. introduced a multifaceted heteroatom doping technique for the direct and continuous fine-tuning of the electronic structure and HER activity of non-noble metals, while retaining their chemical composition [13]. They utilized a pyrolysis method to obtain carbon-supported nickel nanoparticles, subsequently achieving N-P doping using NH_3_ and NaH_2_PO_4_. The outcome, labeled as N-P-Ni, exhibited remarkable performance, manifesting a minimal overpotential of 25.8 mV at 10 mA cm^−2^ (Figure 3a,b). Impressively, the exchange current density (i_0_) of N-P-Ni (1.22 mA cm^−2^) closely paralleled that of commercial Pt/C catalysts (1.3 mA cm^−2^), exceeding pristine Ni by eight-fold (0.154 mA cm^−2^, Figure 3c).

Furthermore, the N-P-Ni displayed a low Tafel slope of 34 mV dec^−1^ (Figure 3d), implying that the rate-determining step had shifted from the Volmer reaction, showcasing enhanced water dissociation kinetics. The electrochemical double-layer capacitance (C_dl_) values of the catalysts were also documented (Figure 3e). Chronoamperometry of N-P-Ni conducted for 50 h at a consistent overpotential of 30 mV showcased minimal alteration in the current response, even after 1000 cycles (Figure 3f).

#### 4.1.3. Ni-M-P (M: Metal) Structure

Hu et al. reported on Ni-Co-P hollow nano-bricks (Ni-Co-P HNBs) for HER [84]. A rapid microwave-assisted technique was employed to synthesize regular cuboid Ag_2_WO_4_ solid nano-bricks (SNBs). Subsequently, these Ag_2_WO_4_ SNBs were combined with Co(NO_3_)_2_, Ni(NO_3_)_2_, and polyvinylpyrrolidone (PVP) to create a core–shell structure in the form of Ag_2_WO_4_@Ni-Co precursor core–shell SNBs. The Ag_2_WO_4_ cores were removed and reacted with NaH_2_PO_2_ to obtain the Ni-Co-P HNBs. The resulting Ni-Co-P HNBs displayed favorable electrocatalytic performance, evidenced by an overpotential of 107 mV at a current density of 10 mA cm^−2^, coupled with a reduced Tafel slope measuring 46 mV dec^−1^. Subsequent chronoamperometry testing revealed a mere 4.3% reduction in current density over a 20 h duration, underscoring the robust durability of the Ni-Co-P HNBs in facilitating the HER within an alkaline environment.

Du et al. devised a sequential method involving hydrothermal, oxidation, and phosphidation steps to craft 3D nest-like ternary NiCoP/carbon cloth (NiCoP/CC) electrocatalysts [85]. Remarkably, the NiCoP/CC composite exhibited outstanding electrocatalytic prowess for the HER, showcasing an extraordinary overpotential of 62 mV at a current density of 10 mA cm^−2^ within an alkaline medium, alongside a Tafel slope measuring 68.2 mV dec^−1^. Even after undergoing 2000 cyclic voltammetry (CV) cycles, the NiCoP/CC experienced a mere 2 mV drop from its initial CV value at the same current density of 10 mA cm^−2^. The superior electrocatalytic activity was further confirmed by the electrochemical double-layer capacitance (C_dl_), recorded at 51.5 mF cm^−2^. Moreover, the NiCoP/CC continued to generate hydrogen and oxygen over 40,000 s at a current density of 100 mA cm^−2^, with the voltage experiencing only a minimal increase to 80 mV after the duration.

Yao et al. successfully prepared mesoporous nanorods of nickel–cobalt–iron–sulfur–phosphorus (NiCoFe-PS), tightly self-supported on a nickel foam substrate (NiCoFe-PS nanorod/NF) [86]. Initially, a NiCoFeZn alloy film was made on the Ni foam via a novel hydrothermal electrodeposition technique. Subsequently, by precisely controlling the potential, selective dealloying was performed, leading to the formation of NiCoFe-PS nanorod/NF. The synthesis process involved the reaction of NiCoFe-PS nanorod/NF with phosphorus and sulfur, facilitated by the decomposition of Na_2_H_2_PO_2_ to yield P. The subsequent sulfidization and phosphorization processes over a span of 1 h resulted in the formation of well-crystallized catalysts. The NiCoFe-PS nanorod/NF electrocatalysts showed remarkable performance metrics, notably an overpotential of merely 97.8 mV at a current density of 10 mA cm^−2^, with a notably low Tafel slope of 51.8 mV dec^−1^.

### 4.2. Co-Based TMPs

#### 4.2.1. Co-P Structure

Cobalt phosphides can cause an “ang-semble effect” in the HER process, such as a metal complex. In addition, Co and P atoms can act as a hydride-acceptor and proton-accepter site to improve electrocatalyst performance [87].

Jiang et al. presented a facile potentiodynamic electrodeposition technique to synthesize Co-P, utilizing common cobalt and phosphorous reagents [49]. The resulting Co-P exhibited exceptional HER performance within an alkaline, demonstrated by an overpotential of 94 mV at a current density of 10 mA cm^−2^, and a Tafel slope measuring 42 mV dec^−1^. While the commercial Pt/C catalysts showcased a minute catalytic onset potential, their larger Tafel slope (108 mV dec^−1^) in comparison to the Co-P film was noteworthy. This led to the Co-P film surpassing the catalytic current density of the commercial Pt/C catalysts beyond −167 mV vs. RHE. Moreover, the Co-P film exhibited enhanced long-term stability. Controlled potential electrolysis (at η = −107 mV) conducted over 24 h demonstrated nearly linear charge accumulation and a consistent current profile throughout the electrolysis process.

Lv et al. detailed the fabrication of well-defined CoP/Co_2_P nanohybrids enveloped within N-doped graphitized carbon shell (CoP/Co_2_P@NC) via a straightforward hydrothermal reaction [88]. Initially, cobalt phosphonate (CoPi) precursors were synthesized through heat treatment, and these precursors were subsequently annealed to yield the CoP/Co_2_P@NC composite. The CoP/Co_2_P@NC-2 variant showcased noteworthy performance, characterized by low overpotentials of 198 mV within an alkaline at a current density of 10 mA cm^−2^. Its Tafel slope, a little higher than that of commercial Pt/C catalysts (82 mV dec^−1^ compared with 73 mV dec^−1^), signified enhanced kinetics for HER. Impressively, after undergoing 1000 cycles of cyclic voltammetry (CV), the polarization curves of CoP/Co_2_P@NC-2 exhibited negligible degradation, underscoring the robust durability of this composite. After long-term HER stability test, it can be confirmed that the core–shell structure without change and clear lattice fringes of CoP/Co2P@NC-2 in HRTEM image showed high structural stability in 1.0 M KOH.

Tabassum et al. introduced a bottom-up synthesis method for encapsulating CoP nanostructures within a B, N co-doped graphene-like carbon framework, referred to as BCN nanotubes (CoP@BCN). This was achieved through a process involving pyrolysis and subsequent phosphorization steps [89]. The CoP@BCN composite showcased remarkable electrochemical prowess in the context of HER across all pH media, sustaining a high level of stability for an extended duration of 8 h. Particularly noteworthy was the CoP@BCN-1 variant, which exhibited the most optimal catalytic HER activities, as evidenced by an overpotential of 215 mV at a current density of 10 mA cm^−2^. Notably, in alkaline, the catalytic current densities were stable at 7.30 mA cm^−2^, experiencing only a minor 10% decline in stability.

#### 4.2.2. Cu-Co-P Structure

Xu et al. showed a successful copper and oxygen dual-doping approach to amplify the count of active sites within CoP (designated as O-Cu-Co-P-2), thereby achieving superior HER performance in alkaline environments [90]. In a 1.0 M KOH, the electrocatalytic effectiveness of O-Cu-Co-P-2 manifested notably lower overpotential (72 mV) at a current density of 10 mA cm^−2^ and a Tafel slope of 57.6 mV dec^−1^, compared to CoP (137 mV and 76.8 mV dec^−1^, respectively). Impedance spectra were acquired at an overpotential of 0.2 V across a frequency range from 100 kHz to 0.1 Hz, using an applied voltage amplitude of 10 mV. Notably, the charge transfer resistance of O-Cu-Co-P-2 was smaller than that of CoP. The quantity of generated hydrogen closely aligned with the theoretical amount, suggesting nearly 100% Faradaic efficiency during the electrolysis process. moreover, accelerated degradation studies underscored the robust durability of O-Cu-Co-P-2 nanowire arrays, as the HER current density exhibited minimal decay even after 5000 cyclic voltammetry (CV) cycles. Additionally, O-Cu-Co-P-2 demonstrated stable overpotentials at both 10 mA cm^−2^ and 50 mA cm^−2^ current densities, maintained over a 24 h duration.

#### 4.2.3. N-Co-P Structure

Luo et al. prepared N-doped Co_2_P supported on carbon cloth (N-Co_2_P/CC) [91]. The Co(OH)F/CC was prepared by heat-treatment, and then a phosphorization process was conducted with NaH_2_PO_2_ to yield the N-Co_2_P/CC (Figure 4a). The N-Co_2_P/CC catalyst demonstrated remarkable catalytic prowess, registering an overpotential of 34 mV at a current density of 10 mA cm^−2^. This performance was similar with the commercial Pt/C catalyst (28 mV), and significantly superior to the Co_2_P/CC (110 mV) and CoP/CC (88 mV) counterparts (as depicted in Figure 4b). Additionally, the Tafel slope of N- Co_2_P/CC was calculated at 51 mV dec^−1^ (Figure 4c). Although this value was a little higher than Pt/CC (42 mV dec^−1^), it was notably lower than the Tafel slopes of Co_2_P/CC (71 mV dec^−1^) and CoP/CC (66 mV dec^−1^). In addition, the almost unchanged LSV curve after 3000 CV cycles with the N-Co_2_P/CC demonstrated its outstanding electrochemical stability (Figure 4d). Furthermore, the EIS outcomes (Figure 4e) illuminated that N-Co_2_P/CC exhibited a notably diminished charge transfer resistance in comparison to pristine Co_2_P/CC. Examining the double-layer capacitance (C_dl_) test curves, it’s evident that N- Co_2_P/CC (141 mF cm^−2^) boasted a larger electrochemical surface area (ECSA) in contrast to CoP/CC (116 mF cm^−2^) and Co_2_P/CC (53 mF cm^−2^), as displayed in Figure 4f. This enlarged ECSA is advantageous in promoting HER performance.

### 4.3. Fe-Based TMPs

#### 4.3.1. Fe-P Structure

Iron phosphides show superb electrical conductivity due to the alloying of metals and phosphorus, and has outstanding electrochemical properties due to its excellent catalytic activity, corrosion resistance, and stability compared to other TMPs [1].

Huang et al. employed a straightforward and gentle sol–gel technique to produce a self-supported electrode comprised of mesoporous FeP (designated as meso-FeP/CC) [92]. The catalytic activity of meso-FeP/CC for HER was highly impressive, prompting a current density of 10 mA  cm^−2^ with an overpotential of 84 mV. Notably, this overpotential was 61 mV and 72 mV lower than that of FeP/CC and meso-FeP, respectively. Furthermore, the meso-FeP/CC boasted the smallest Tafel slope among other FeP-based electrodes, measuring at 60 mV  dec^−1^. This indicated swifter kinetics for the HER process. Remarkably, the electrode also demonstrated robust stability during a 20 h chronoamperometric assessment conducted within an alkaline medium. After HER measurement, there was no significant change in the XRD and XPS results, and original mesoporous morphology of meso-FeP/CC was maintained in TEM, confirming excellent durability.

Shi et al. introduced an innovative approach involving the electrodeposition of transition metal phosphide (FeP) cubes with nanoporous structures onto carbon paper (CP) as an effective catalyst for the HER [79]. This involved the rapid fabrication of nanoporous FeP cubes on CP, referred to as NPC FeP/CP, through electrodeposition followed by acid-etching (Figure 5a). To assess its effectiveness, HER polarization curves were generated for Fe/FeOOH, Fe/FeOOH/FeP, and FeP in a 1.0 M KOH, with commercial Pt/C catalysts serving as a reference for comparison (Figure 5b). The commercial Pt/C catalysts showed outstanding HER activity, marked by a low overpotential of 29 mV at a current density of 10 mA cm^−2^. A small Tafel slope of 42.70 mV dec^−1^ was measured through linear fitting of the polarization curves using the Tafel equation (Figure 5c). Following the acid-etching process that eliminated Fe and FeOOH species, the resulting NPC FeP/CP manifested an overpotential of 140 mV at a current density of 10 mA cm^−2^, coupled with a notably diminished Tafel slope of 61.92 mV dec^−1^. This reduction in the Tafel slope indicated favorable HER kinetics for NPC FeP, operating through the Volmer–Heyrovsky mechanism within alkaline environments. Remarkably, all the FeP/CP samples exhibited commendable HER activity in 1.0 M KOH, with their overpotentials at a current density of 10 mA cm^−2^ registering at 249, 181, 140, 152, and 170 mV for FeP5min, FeP15min, FeP30min, FeP45min, and FeP60min, respectively. This data showed that the optimal HER performance was achieved by NPC FeP30min (Figure 5d). Indeed, the NPC FeP30min/CP configuration demonstrated the most favorable Tafel slope of 61.92 mV dec^−1^ among all the FeP samples (Figure 5e). As shown in Figure 5f, the multi-current HER processes on NPC FeP were observed in both acid and alkaline electrolytes. Upon gradually increasing the potential by 0.01 V every 500 s, the corresponding current density underwent a continuous enhancement before swiftly stabilizing. This succession of rapid and cyclic staircase-like steps vividly illustrated the advantageous mass transport characteristics, encompassing the inward diffusion of OH^−^ or H^+^ ions and the outward diffusion of H_2_ bubbles, as well as the exceptional conductivity of NPC FeP.

#### 4.3.2. M-Fe-P (M = Metal) Structure

Lu et al. detailed the creation of uniform hollow nanorods composed of Ni-doped FeP (NFP/C) nanocrystals hybridized with carbon, designed as electrocatalysts for HER [93]. The incorporation of Ni led to abundant active sites and an improved capability for mass and charge transport, rendering the optimized Ni-doped FeP/C hollow nanorods as exceptional catalysts for the HER in a 1.0 M KOH environment. These doped hollow nanorods outperformed the FeP/C sample, displaying superior HER activities characterized by a smaller overpotential and Tafel slope (measured at 72 mV dec^−1^). Furthermore, these Ni-doped FeP/C hollow nanorods demonstrated heightened properties, notably showcasing a smaller overpotential and a higher exchange current density (*J*_0_ = 0.481 mA cm^−2^).

Yang et al. prepared porous nanospindle composed of a carbon-encapsulated MoO_2_-FeP (MoO_2_-FeP@C) [94]. The fabrication process of MoO_2_-FeP@C involved a PMo_12_@Fe complex (Fe)/FeOOH precursor, which was transformed using the POMOF method with FeOOH serving as a self-sacrificial template [95,96]. The MoO_2_-FeP@C showcased an elevated catalytic activity, demonstrating overpotentials of 103 mV and 190 mV at current densities of 10 mA cm^−2^ and 100 mA cm^−2^, respectively. These values were notably lower than those observed for FeP@C (155 mV and 258 mV), MoO_2_@C (182 mV and 274 mV), and MoO_2_-FeP (128 mV and 214 mV). The Tafel slope for MoO_2_-FeP@C measured 48 mV dec^−1^, presenting a lower value than that of FeP@C (55 mV dec^−1^) and MoO_2_@C (60 mV dec^−1^). This finding highlighted the accelerated reaction kinetics exhibited by MoO_2_-FeP@C.

#### 4.3.3. Fe-P/M-P (M = Metal) Structure

Qin et al. presented their research on well-defined FeP–CoMoP hierarchical nanostructures (FeP–CoMoP HNSs/CC) used as electrocatalysts for the HER in alkaline [97]. Their study included comparisons with Co NRs/CC, Co-Mo NTs/CC, and Fe-Co-Mo HNSs/CC, all of which showed negligible performance due to their intrinsic low conductivity and limited active sites. However, the situation changed significantly after phosphorization. The FeP–CoMoP HNSs/CC displayed an extraordinarily low overpotential of 33 mV at a current density of 10 mA cm^−2^, a remarkable improvement over the CoP NRs/CC (93 mV) and CoMoP NTs/CC (72 mV). Additionally, in terms of Tafel slope, the FeP–CoMoP HNSs/CC (51 mV dec^−1^) outperformed the CoMoP NTs/CC (87 mV dec^−1^) and CoP NRs/CC (102 mV dec^−1^), indicating significantly faster HER kinetics for FeP–CoMoP HNSs/CC.

### 4.4. Mo-Based TMPs

#### 4.4.1. Mo-P Structure

Molybdenum phosphide is a catalyst mainly used in hydrodesulfurization (HDS) reaction. MoP has an excellent HER catalytic performance because HDS and HER depend on the reversible combination of catalyst and hydrogen [98].

Zhang et al. prepared MoP/CNTs with small-sized and well-crystallized MoP nanoparticles coated uniformly on the sidewalls of carbon nanotubes (CNTs, multiwall carbon nanotubes) for HER [99]. The conductivity, HER activity, and stability of the MoP/CNTs were higher than MoS_2_/CNTs, MoN_x_/CNTs, and MoO_x_/CNTs. Thus, in 1.0 M KOH, the MoP/CNTs showed an even higher HER activity (overpotential of 86 mV at a current density of 10 mA cm^−2^), and a much higher stability (27 mV decay at a current density of 10 mA cm^−2^ for 40 h). Post HER measurement, it can be seen that the MoP/CNT-700 maintained its original structure by SEM, TEM, XRD, indicating high stability in morphology and structure of the hybrids.

Wu et al. devised a straightforward and scalable procedure to produce molybdenum phosphide coupled with reduced graphene oxide (MoP–RGO) [100]. In a 1.0 M KOH, MoP–RGO-800 displayed overpotentials of 238 mV at a current density of 10 mA cm^−2^, which were comparatively lower than MoP–RGO-700 (424 mV) and MoP–RGO-900 (247 mV). This highlights the significance of pyrolysis temperature in fine-tuning the electrocatalytic performance for the HER. Furthermore, the introduction of NaCl as a template was employed to enhance the specific surface area of the catalysts, enabling the creation of pores. The electrocatalytic performance of MoP–RGO-800 was assessed in 1.0 M KOH, showcasing an overpotential of 238 mV at a current density of 10 mA cm^−2^. This value was comparatively lower than that of MoP–RGO-700 (424 mV) and MoP–RGO-900 (247 mV), highlighting the impact of pyrolysis temperature as a crucial factor for tuning electrocatalytic performance in the HER. To further enhance specific surface area, NaCl was utilized as a template to create pores within the catalysts [101]. With the addition of 0.5 g of NaCl, the overpotential at a current density of 10 mA cm^−2^ decreased from 238 to 183 mV. MoP–RGO-0.5 exhibited the lowest Tafel slope (69 mV dec^−1^) compared with MoP–RGO (74 mV dec^−1^), MoP–RGO-0.2 (95 mV dec^−1^), and MoP–RGO-0.8 (134 mV dec^−1^), approaching the Tafel slope of commercial Pt/C catalysts (53 mV dec^−1^).

Song et al. conducted a study on N-doped, defect-containing, carbon-dot (CDs)-loaded molybdenum phosphide (MoP/CDs) nanoparticles by utilizing CDs with varying N contents [102]. To increase the N content in the CDs, the electrocatalytic activity for HER was progressively enhanced. The overpotential of MoP/0.5CM–CDs1100 at a current density of 10 mA cm^−2^ was 70 mV, although this was higher than the commercial Pt/C catalysts (32 mV) (Figure 6a). The Tafel slope of MoP/0.5CM–CDs1100 (77.49 mV dec^−1^) was a little higher than the commercial Pt/C catalysts (54.45 mV dec^−1^), yet lower than MoP/0.5CE-CDs1100 (98.61 mV dec^−1^) and MoP/0.5CA-CDs1100 (102.57 mV dec^−1^), suggesting the dominance of the Volmer–Heyrovsky mechanism in 1.0 M KOH (Figure 6b). EIS results revealed that MoP/0.5CM–CDs1100 displayed a smaller charge transfer resistance than MoP/0.5CE-CDs1100 or MoP/0.5CA-CDs1100 (Figure 6c). Stability testing of MoP/0.5CM–CDs1100 through 1000 CV cycles demonstrated minimal change in overpotential at a current density of 10 mA cm^−2^ (Figure 6d). The trend in overpotentials was ranked as follows: MoP/0.5CM–CDs1100 (70 mV) < MoP/0.5CM–CDs1000 (93 mV) < MoP/0.5CM–CDs900 (134 mV) < MoP/0.5CM–CDs800 (296 mV), indicating that activity increased with higher annealing temperatures (Figure 6e). Notably, employing CM–CDs at half the reference mass yielded the optimal (i.e., smallest) overpotential and Tafel slope (70 mV at a current density of 10 mA cm^−2^ and 77.49 mV dec^−1^, respectively) among the tested catalysts with different masses of CM–CDs (Figure 6f).

#### 4.4.2. Mo-Ni-P Structure

Sun et al. presented a study on the utilization of Mo-doped Ni_2_P nanowires on Ni foam (Mo-Ni_2_P NWs/NF) as a proficient and enduring electrocatalyst for HER [103]. The Mo-Ni_2_P/NWs on conductive Ni foam were synthesized by converting NiMoO_4_ nanowire arrays through a topotactic phosphidation reaction. The excellent HER performance of the Mo-Ni_2_P NWs/NF can be ascribed to the inclusion of Mo into Ni_2_P and the synergistic effects between the components. The Mo-Ni_2_P NWs/NF exhibited a small overpotential of 78 mV at a current density of 10 mA cm^−2^ and maintained long-term stability (>24 h) in 1.0 M KOH. This performance surpasses that of many Co- and Fe-doped binary transition metal phosphides. Even after 2500 sweeps, the polarization curve displayed minimal deviation from the initial curve, indicating minimal corrosion of the electrocatalyst in the alkaline environment.

#### 4.4.3. Mo-P/Ni-P Structure

Du et al. prepared hierarchical MoP/Ni_2_P heterostructures on a 3D Ni foam (MoP/Ni_2_P/NF) and evaluated its potential as an effective bifunctional electrocatalyst for water splitting [104]. In comparison to both the precursor and pure Ni foam, the MoP/Ni_2_P/NF exhibited a remarkable enhancement in its catalytic performance for HER, demonstrating low overpotentials of 75 mV and 191 mV at current densities of 10 mA cm^−2^ and 100 mA cm^−2^, respectively. This underscores the superior efficiency of the MoP/Ni_2_P/NF as a cathode for HER. The Tafel slope was measured to be 100.2 mV dec^−1^, which aligns with the typical range of 40 to 120 mV dec^−1^, implying that the HER process likely followed the Volmer–Heyrovsky mechanism on the MoP/Ni_2_P/NF surface. Through stability testing in 1.0 M KOH under a constant overpotential of 125 mV (~22 mA cm^−2^), the MoP/Ni_2_P/NF exhibited negligible deterioration in current density over 24 h, illustrating the excellent electrochemical durability of the hierarchical MoP/Ni_2_P/NF heterostructures for HER in alkaline conditions.

### 4.5. Others

Pu et al. conducted research on the synthesis of tungsten phosphide nanorod arrays on carbon cloth (WP NAs/CC) through a two-step process involving the hydrothermal growth of WO_3_ nanorod arrays on carbon cloth (WO_3_ NAs/CC) followed by phosphidation to chemically convert the WO_3_ NAs/CC precursor into WP NAs/CC [105]. The electrocatalytic performance of WP NAs/CC was evaluated, yielding overpotentials of 150 mV and 271 mV at current densities of 10 mA cm^–2^ and 100 mA cm^–2^, respectively. Additionally, the Tafel slope of 102 mV dec^–1^ was observed in the potential region of η = 120–250 mV. These results indicate the potential of WP NAs/CC as an effective electrocatalyst for the HER in alkaline.

Hou et al. presented a study involving the growth of cedarlike semimetallic Cu_3_P nanoarrays directly on a 3D copper foam (CF) substrate [106]. The significant roughness factor (RF) of the Cu_3_P nanoarrays contributed to a highly electrochemically active surface area, while the semimetallic nature of the Cu_3_P core facilitated efficient charge transfer. In terms of electrocatalytic activity, the Cu_3_P/CF, CF, and commercial Pt/C catalysts exhibited overpotentials of 222 mV, 541 mV, and 57 mV at a current density of 10 mA cm^–2^, respectively. In addition, the Tafel slope for Cu_3_P/CF was measured at 148 mV dec^–1^, which is similar to the reported value for CoP on carbon cloth in alkaline and lower than CF (184 mV dec^–1^).

**Table 2 nanomaterials-13-02613-t002:** Comparison of the performance of HER electrocatalysts in alkaline media.

Electrocatalyst	Overpotential (mV) at a Current Density of 10 mA cm^−2^	Tafel Slope (mV dec^−1^)	Ref.
Ni_2_P/Ni composite	41	50	[82]
Ni-FeP/TiN/CC	75	73	[107]
Ni/Ni_2_P@3DNSC	92	65	[108]
NiCoFe-PS nanorod/NF	97.8	51.8	[86]
NSP-Ni_3_FeN/NF	45	75	[109]
Ni_2_P/Ni_0.96_S/NF	72	149	[110]
NiCoP/CC	62	68.2	[85]
Ni_2_P/MoO_2_/NF HNRs	34	45.8	[111]
sc-Ni_2_P^δ−^/NiHO	60	75	[112]
Ni_2_P-Ni_3_S_2_ HNAs/NF	80	65	[113]
NiMoP	400	163	[114]
NiMnP	490	238	[114]
NiFeP	690	116	[114]
NiCoP	530	116	[114]
Ni_2_P	710	103	[114]
N-NiCoP/NCF	78	83.2	[115]
v-Ni_12_P_5_	27.7	30.88	[116]
Ni_2_P-NiP_2_ HNPs/NF	59.7	58.8	[117]
Ni_2_P/rGO	142	58	[83]
Ni-Co-P HNBs	107	46	[84]
Ni_2_P/Ni/NF	98	72	[118]
Ni_5_P_4_	47	56	[119]
NiCo_2_P_x_ NW	58	34.3	[120]
Ni_2_P-NiCoP@NCCs	116	79	[121]
(Ni_0.33_Fe_0.67_)_2_P NS	84		[122]
NiFe LDH@NiCoP/NF	120	88.2	[123]
Ni(OH)_2_-Fe_2_P/TM	76	105	[124]
R-NiZnP/NF	50	53	[125]
N-P-Ni	25.8	34	[13]
N–NiCoP NWs/CFP	105.1	59.8	[126]
Co-P	94	42	[49]
CoP/CC	209	129	[30]
CoP/Co_2_P@NC	198	82	[88]
CoP@BCN	215		[89]
O, Cu-CoP-2	74	57.7	[90]
S-CoP@NF	109	79	[127]
Zn_0.075_, S-Co_0.925_P NRCs/CP	37	41.5	[128]
CoFeP	177	72	[129]
Co_3_S_4_/MoS_2_/Ni_2_P	178	98	[130]
Zn_0.08_Co_0.92_P/TM	67		[131]
Al-CoP/CC	38	45	[132]
Al-CoP/NF	66	94	[133]
N-Co_2_P/CC	34	51	[91]
C-(Fe-Ni)P@PC/(Ni-Co)P@CC	142	98	[134]
CoP-CeO_2_/Ti	43	45	[135]
Co_0.75_Fe_0.25_P	209	55.5	[136]
FeP (NPC FeP/CP)	140	61.92	[79]
FeP NAs/CC	218	146	[137]
meso-FeP/CC	84	60	[92]
FeP-CoMoP	33	51	[97]
NFP/C	95	72	[93]
MoO_2_-FeP@C	103	48	[94]
MoP/CNT	86	73	[99]
MoP_2_ NS/CC	67	70	[138]
MoP-RGO	152	69	[100]
MoP/MWCNTs	155	56.8	[139]
Mo-Ni_2_P NWs/NF	78	109	[103]
MoP/Ni_2_P/NF	75	100.2	[104]
MoP/CDs	70	77.49	[102]
A-MoP@PC	67	40	[140]
C-WP/W	133	70.1	[141]
Cu_3_P/CF	222	148	[106]
WP NAs/CC	150	102	[105]

## 5. Conclusions and Perspectives

The inventive development and preparation of non-noble metal HER electrocatalysts are crucial because of the strong correlation between material characteristics, such as morphology and structure [142]. Despite considerable advancements of high-performance HER electrocatalysts, further endeavors are required to enable their practical use in commercial settings for sustainable hydrogen production. So, future research should integrate theoretical and experimental investigations to anticipate and validate the electrocatalytic efficiency of TMPs (Figure 7). Gaining a comprehensive understanding of the electrocatalyst’s structural evolution during electrolysis holds immense importance, as the reconfigured surface plays a pivotal role in determining the sustained performance over time [143]. Accurate synthesis control leading to meticulously crafted surfaces featuring minimal defects and targeted crystalline structures or facets holds substantial significance in constructing theoretical models. This is crucial due to the potential impact of hydrogen adsorption on interatomic forces at the surface of catalysts, subsequently influencing the performance of hydrogen evolution [29,144].

Investigating the molecular-level interfacial structure of catalysts during electrochemical hydrogen evolution stands as a significant challenge. Gaining a comprehensive understanding of the catalytic behavior of transition metal phosphide (TMP)-based catalysts in HER is crucial for exploring the catalysts’ structural changes and the conversion of reactants, intermediates, and products. Given the dynamic nature of transient states and the inevitable oxidation of catalysts, as well as the presence of unstable intermediate species, conventional ex situ approaches may fall short in capturing the evolving surface and providing precise insights into intermediate transformations.

To address this, employing in situ characterization methods, such as in situ Raman measurements and in situ X-ray absorption spectroscopy, holds promise in revealing interface signals of non-noble metal HER electrocatalysts. These techniques offer the potential to observe catalysts’ behaviors in real-time, mitigating challenges associated with transient states and oxidation effects. Attaining these research objectives, which illuminate the true active sites and reaction mechanisms, could pave the way for the development of advanced electrocatalysts [29,62].

Density Functional Theory (DFT) calculations have become an increasingly valuable method for gaining insights into performance predictions and catalyst design, and their power is continuously growing [29,62]. Understanding how the crystal structure, chemical composition, and electronic state of TMPs influence HER performance, as well as elucidating the effects of doping with other elements and the intricate interactions between coupled materials and TMPs, presents challenges. Nevertheless, DFT calculations offer a promising avenue for unraveling these complexities. By revealing the modulations in electronic structure and interactions among reactive species and surface structures, DFT can shed light on the evolution of intermediates. Moreover, DFT calculations enable the prediction of optimal crystal structures and chemical compositions, facilitating the quest for improved catalyst designs. Acquiring these fundamental understandings will provide us with profound insights into the underlying mechanisms governing the functionality and enduring stability of TMPs throughout the process of electrolysis.

Hydrogen evolution reaction (HER) will become an integral part of sustainable energy in the future because hydrogen is considered the most promising candidate for clean fuel used in fuel cell technologies. Therefore, it is important to achieve economical hydrogen production. Precious metals like Pt, Ru, Pd, Ir, etc. have been considered for HER electrocatalysts, but they are expensive and scarce and have poor stability. Thus, research on finding inexpensive and practical electrocatalysts should be actively conducted. This review provides an inclusive overview of the latest development in electrocatalysts for the HER, particularly highlighting the potential of cost-effective TMPs. Notably, TMPs have exhibited exceptional activity, stability, and affordability across a broad pH spectrum. TMPs can be prepared easily with phosphorus precursors through various reactions. The P content of TMPs has a crucial role in the performance for the HER because P atoms affect electrical conductivity, reactivity and stability. TMPs possess a notable electrical conductivity, a characteristic that proves advantageous in expediting charge transfer. Additionally, the Gibbs free energy associated with the HER intermediates falls within a moderate range, signifying a pronounced intrinsic activity conducive to efficient hydrogen evolution. In other words, due to its highly active properties, TMPs show great potential to replace expensive precious metals as HER electrocatalysts. Nonetheless, there remains a need for further enhancement in the electrocatalytic performance of TMP-based materials, aiming to surpass the capabilities of commercial Pt/C catalysts. Therefore, even though the development of catalysts has made significant progress, further advances are needed for sustainable hydrogen production and commercial applications.

## Figures and Tables

**Figure 1 nanomaterials-13-02613-f001:**
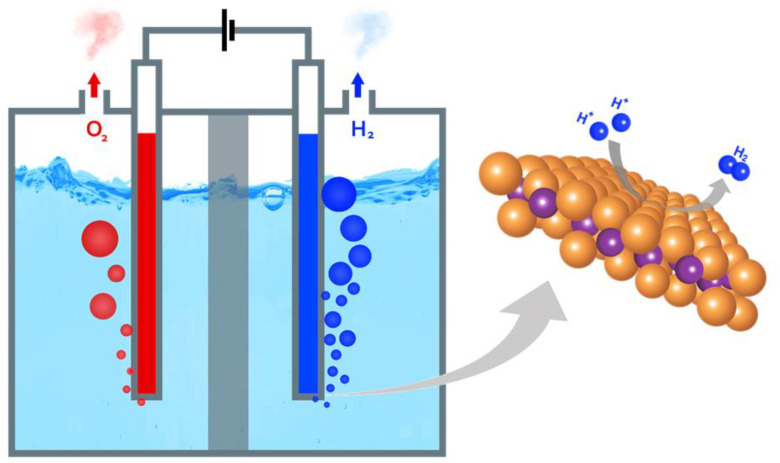
Hydrogen evolution reaction through water splitting.

**Figure 2 nanomaterials-13-02613-f002:**
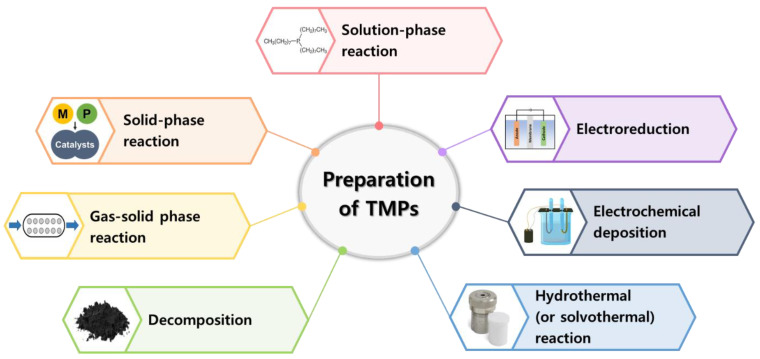
Preparation of transition-metal phosphides (TMPs).

**Figure 3 nanomaterials-13-02613-f003:**
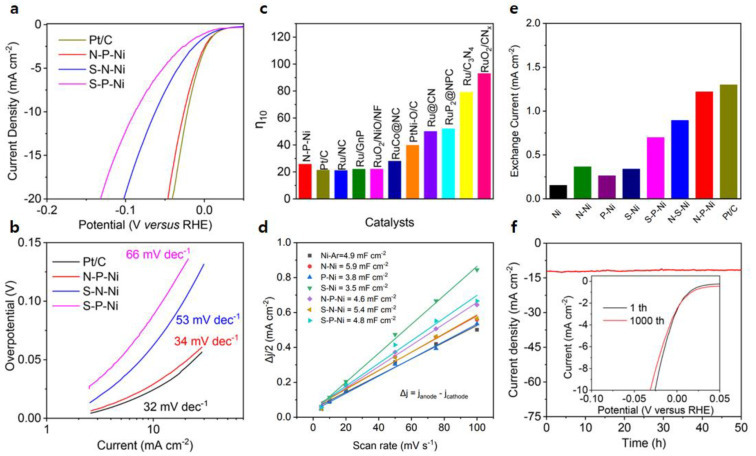
Electrochemical performance for the HER: (**a**) LSV curves; (**b**) Tafel slopes; (**c**–**e**) Comparison of the overpotential (η_10_) at a current density of 10 mA cm^−2^, exchange current density (i_0_), and electrochemical double layer capacitance (C_dl_); (**f**) Chronoamperometric curves (Inset: LSV of N-P-Ni in a long-term stability test). Adapted with permission from Ref. [13].

**Figure 4 nanomaterials-13-02613-f004:**
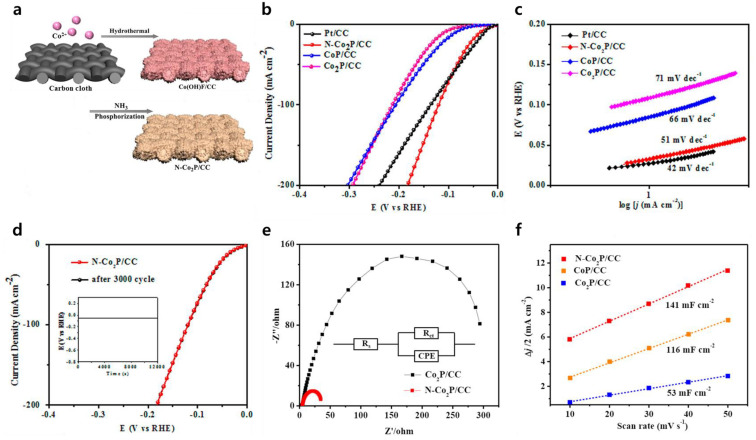
(**a**) Preparation of the N-Co_2_P/CC. Electrochemical performance for the HER: (**b**) Polarized curves; (**c**) Tafel slopes; (**d**) Polarized curves of the N-Co_2_P/CC at the first cycle and after 3000 cycles (Inset: the chronopotentiometric curve with a constant current density of 10 mA cm^−2^ for 120,000 s); (**e**) Nyquist plots; (**f**) Electrochemical double-layer capacitance (C_dl_). Adapted with permission from Ref. [91].

**Figure 5 nanomaterials-13-02613-f005:**
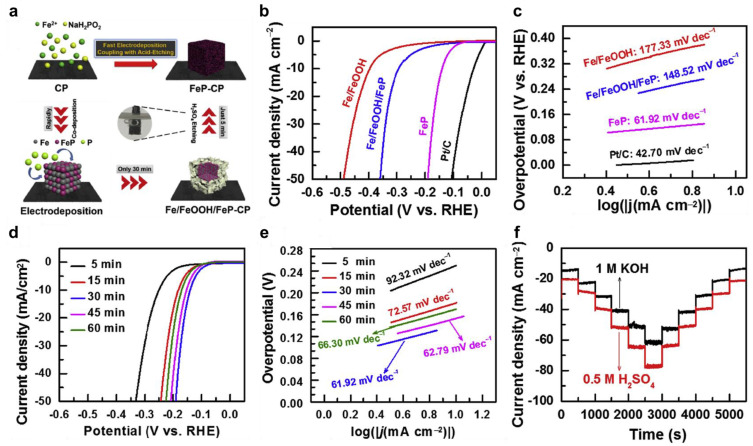
(**a**) Schematic illustration for the preparation process of FeP-CP. Electrochemical performance for the HER: (**b**) Polarization curves; (**c**) Tafel plots; (**d**) Polarization curves of the FeP samples prepared with different electrodeposition times; (**e**) Tafel plots of the FeP samples prepared with different electrodeposition times; (**f**) Multi-current HER processes with a step of −0.01 V every 500 s for the FeP samples in 1.0 M KOH with potential increases from −0.17 to −0.22 V (vs. RHE) and in 0.5 M H_2_SO_4_ with potential increases from −0.12 to −0.17 V (vs. RHE), respectively. Adapted with permission from Ref. [79].

**Figure 6 nanomaterials-13-02613-f006:**
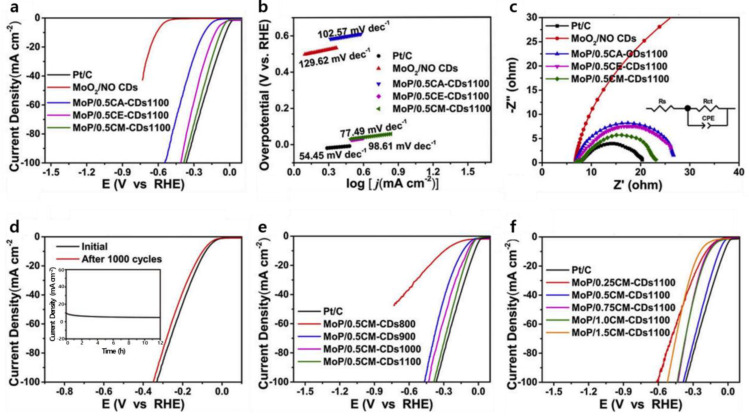
Electrochemical performance for the HER: (**a**) Polarization curves; (**b**) Corresponding Tafel slopes; (**c**) Nyquist curves; (**d**) Durability test for the MoP/0.5CM–CDs1100 in 1.0 M KOH (Inset: the current-time (*i*–*t*) curves at a constant overpotential for 12 h); (**e**) Polarization curves for the MoP/CM–CDs annealed at different temperatures; (**f**) Polarization curves of the MoP/CM–CDs1100 prepared by adding different CM–CDs ratios. Adapted with permission from Ref. [102].

**Figure 7 nanomaterials-13-02613-f007:**
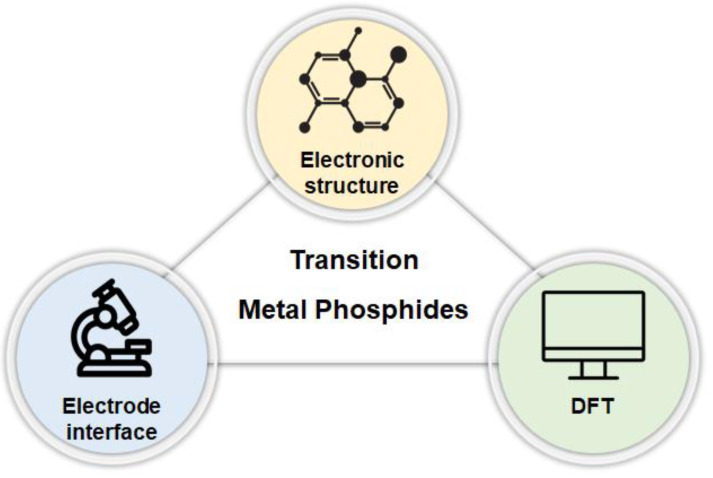
Key points for determining the HER performance of TMPs.

**Table 1 nanomaterials-13-02613-t001:** Mechanism of the HER.

Electrolyte	Reaction Pathway	Equation
**Alkaline electrolyte**	**Volmer** H_2_O + e^−^ → H_ads_ + OH^−^ **Heyrovsky** H_ads_ + H_2_O + e^−^ → H_2_ + OH^−^ **Tafel** H_ads_ + H_ads_ → H_2_ (g)	2H_2_O + 2e^−^ → H_2_ + 2OH^−^
**Acidic electrolyte**	**Volmer** H_3_O^+^ + e^−^ → H_ads_ **Heyrovsky** H_ads_ + H_3_O^+^ + e^−^ → H_2_ **Tafel** H_ads_ + H_ads_ → H_2_ (g)	2H^+^ + 2e^−^ → H_2_

## Data Availability

Data presented in this study are available on request from the corresponding author.

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
