# Peer review of "Recent Tendency on Transition-Metal Phosphide Electrocatalysts for the Hydrogen Evolution Reaction in Alkaline Media"

_nanomaterials, 2023, doi:10.3390/nano13182613_

Round 1

Reviewer 1 Report

This article entitled ' Recent Tendency on Transition-Metal Phosphide Electrocatalysts  for the Hydrogen Evolution Reaction in Alkaline Media ' is a comprehensive article on Transition-Metal Phosphide Electrocatalysts  for the HER catalysis. This review is well organized and well detailed. It gives an interesting overview on the preparation and the performance of such original materials in the field of energy and hydrogen production.

I just suggest to combine the section ‘5-Future plan’ with the conclusion to give a conclusive section ‘Conclusion and Perspectives.

 I recommend acceptance of this article

Author Response

September 19, 2023

Dear

We appreciate the reviewers’ reports and editorial comments on our manuscript entitled: “Recent Tendency on Transition-Metal Phosphide Electrocatalysts for the Hydrogen Evolution Reaction in Alkaline Media” (No. Nanomaterials-2616646). We have carefully examined the reviewers’ concerns and suggestions. We are attaching the revised manuscript that includes amendments made in accordance with the reviewers’ suggestions, along with our point-to-point responses.

As you can see, we have clarified the referees’ recommendations. The modifications made are highlighted in yellow for the reviewer #1, green for the reviewer #2 and blue for the reviewer #3. Now we believe that the revised manuscript is in a good shape and hope that the paper is now acceptable for publication in. Nanomaterials.

Once again, thank you very much for your kind consideration and editorial effort. We look forward to hearing from you again soon.

Yours sincerely

In-Yup Jeon

Reviewer 2 Report

Comments to the Authors

In this review, Yoon and co-authors summarized the recent advances about transition-metal phosphides (TMPs)-based electrocatalysis for highly efficient hydrogen evolution reactions (HER). The analyzed the mechanism behind the HER electrocatalysis in both acidic and alkaline conditions, and then the preparation methods of TMPs were reviewed. Finally, they introduced the HER performance and the enhanced mechanism of TMPs by classifying different TMPs. This review has been well orginized, and I recommend its publication after addressing the following issues.

1. The authors said that HER catalysts using abundant elements actively have been intensively developed because the alkaline medium serves as a promising platform for materials that might not be as appealing in acidic conditions due to their limited stability. However, a large number of transition-metal-based HER catalysts have shown excellent performance in acidic media (J. Mater. Chem. A 2021, 9, 23574-23581; Int. J. Hydrogen Energy 2023, 48, 31101-31109; J. Mater. Chem. A 2022, 10, 24927-24937, etc.).

2. Tafel values of HER electrocatalysis should be mentioned in Section 2, because they determine the kinetics of HER electrocatalysis.

3. In Section 3, several preparation examples should be taken to demonstrate the feasibility of every strategy, such as Solution-phase reactions (Nanoscale 2016, 8, 16187-16191), Gas-solid phase reaction (J. Mater. Chem. A 2021, 9, 8561-8567), Hydrothermal (or solvothermal) reaction (Appl. Catal. B 2023, 339, 123136), etc.

4. Before introducing each metal phosphide, the advantages of TMPs on HER electrocatalysis should be included, such as Ni-P, Co-P.

5. After understanding the HER electrocatalysis of NiP, the authors should have some transitional language to describe other Ni-P-based structures, such as doping effect (Nano Energy 2023, 115, 108679) and interface engineering (Chem. Eng. J. 2023, 453, 139796).

6. Several TMPs-based HER electrocatalysts must be mentioned due to their significance in this field (Sci. China Mater. 2020, 6, 240-248; Nano-Micro Lett. 2021, 13, 215; Appl. Catal. B 2023, 328, 122487; Nano Res. 2023, 16, 8765-8772; Nanoscale 2021, 13, 14179-14185; Chem. Eur. J. 2020, 26, 13305-13310).

Author Response

(The authors gave the same response as above.)

Reviewer 3 Report

In this review, the authors focus on recently developed transition metal phosphides for alkaline HER, where the challenges are also discussed. However, several important issues should be solved before publication.  

1.     In the introduction part, the authors should at least discuss about current requirement of hydrogen energy and hydrogen economics to highlight the importance of green hydrogen production, which will help improve the readability. The authors can refer to this work (DOI: 10.1002/adma.202305074).

2.     Some review papers about transition metal phosphides for alkaline HER have been published previously. The authors should compare and conclude from these papers, for example this work (DOI: 10.1016/j.cej.2023.141674).

3.     Multi-phase transition metal phosphides are an important catalyst family applied for alkaline HER. The authors should at least discuss about these catalysts.

4.     The structural reconstruction of transition metal phosphides into metals during/after alkaline HER should be discussed and included in the further efforts.

Minor editing of English language required

Author Response

(The authors gave the same response as above.)
